Automated composition of Galician Xota—tuning RNN-based composers for specific musical styles using deep Q-learning

Mira Rodrigo 1 rs2517@imperial.ac.uk
http://orcid.org/0000-0001-5234-1497 Coutinho Eduardo 2 e.coutinho@liverpool.ac.uk
Parada-Cabaleiro Emilia 3
Schuller Björn W. 1 4
1 GLAM—Group on Language Audio & Music, Department of Computing, Imperial College London , London , United Kingdom
2 Applied Music Research Lab, Department of Music, University of Liverpool , Liverpool , United Kingdom
3 Department of Music Pedagogy, Nuremberg University of Music , Germany
4 ZD.B Chair of Embedded Intelligence for Health Care and Wellbeing, Universität Augsburg , Augsburg , Germany
Stowell Dan
Electronic publication date: 2023 May 15
Publication date: 2023
Volume: 9
Electronic Location ID: e1356
Received 2022 Jun 27; Accepted 2023 Mar 30
Copyright: © 2023 Mira et al.
Copyright year: 2023
Copyright holder: Mira et al.
License: This is an open access article distributed under the terms of the Creative Commons Attribution License, which permits unrestricted use, distribution, reproduction and adaptation in any medium and for any purpose provided that it is properly attributed. For attribution, the original author(s), title, publication source (PeerJ Computer Science) and either DOI or URL of the article must be cited.
License URL: https://creativecommons.org/licenses/by/4.0/

Keywords: Automated music composition, Galician Xota, Magenta, RL Tuner, Deep Q-Learning

Funding: The authors received no funding for this work.

==============================
Music composition is a complex field that is difficult to automate because the computational definition of what is good or aesthetically pleasing is vague and subjective. Many neural network-based methods have been applied in the past, but they lack consistency and in most cases, their outputs fail to impress. The most common issues include excessive repetition and a lack of style and structure, which are hallmarks of artificial compositions. In this project, we build on a model created by Magenta—the RL Tuner—extending it to emulate a specific musical genre—the Galician Xota. To do this, we design a new rule-set containing rules that the composition should follow to adhere to this style. We then implement them using reward functions, which are used to train the Deep Q Network that will be used to generate the pieces. After extensive experimentation, we achieve an implementation of our rule-set that effectively enforces each rule on the generated compositions, and outline a solid research methodology for future researchers looking to use this architecture. Finally, we propose some promising future work regarding further applications for this model and improvements to the experimental procedure.

Introduction

Since the inception of artificial intelligence (AI) there have been many attempts to automate creativity in various areas (Al-Rifaie & Bishop, 2013; Boden, 1997; DiPaola & McCaig, 2016), and one of the most elusive fields has always been music composition. Indeed, algorithmic processes for generating musical creativity have been proposed for various decades, but they cannot be considered truly intelligent since often they operate on a list of commands or instructions devised by human beings instead of learning independently (Doornbusch, 2010). With the advent of deep learning (DL) (LeCun, Bengio & Hinton, 2015) and its exponential increase in popularity over the past few years, many promising methods have been proposed for composing music. These include recurrent neural networks (RNN, typically using long short-term memory cells—LSTM) (Brunner et al., 2017; Chu, Urtasun & Fidler, 2016; Eck & Schmidhuber, 2002), generative adversarial networks (GANs) (Dong et al., 2018; Mogren, 2016; Yang, Chou & Yang, 2017; Yu et al., 2017), Variational Autoencoders (VAEs) (Sabathe, Coutinho & Schuller, 2017) and even Deep Q Networks (DQNs) (Jaques et al., 2016). Despite these developments, most attempts at automated music composition so far have struggled to produce relevant musical outputs, and one pressing issue which is consistent among many different works in this field is the lack of style and structure in the generated musical pieces (Brunner et al., 2017; Chen & Miikkulainen, 2001; Chu, Urtasun & Fidler, 2016). Furthermore, a common concluding remark in most studies in this field is that artificially generated compositions typically appear to wander without any distinguishable musical idea, lack style, and exhibit no structural organisation. Although these aspects are central to music composition, only a small fraction of recent works focus on the issues of style (Jin et al., 2020) and structure (Chen et al., 2018; Dai et al., 2021; Zou et al., 2022). Instead, most works choose to focus on other developments such as polyphony (Dong et al., 2018) and composing with raw audio (van den Oord et al., 2016). Recently, large transformer-based architectures have also been proposed to generate music with long-term structure (Hawthorne et al., 2022; Huang et al., 2019), but require large datasets and substantial computational resources to be trained effectively. Furthermore, these works focus their evaluation on the model’s validation loss (negative log-likelihood), rather than proposing any specific metrics to evaluate structure or style objectively.

A prevalent recent work that tackles the issue of style and structure directly is the RL Tuner (Jaques et al., 2016), proposed by researchers working on Google Magenta (Magenta, 2019a), a research project applying machine learning to creative processes. Reinforcement Learning (RL) is an area of machine learning which does not frequently coincide with music generation. However, this project introduces Double Deep Q Learning (Mnih et al., 2013; van Hasselt, Guez & Silver, 2015) as a way to shape and tune RNN-based “composers” with a novel architecture. In this environment, the agent is the composer, the state is the composition so far and the action is the next note. The reward is partly the output of an LSTM network trained with real music and partly determined by a set of music theory rules which aim to make the compositions adhere to some basic standards of Western music composition. This state space is then explored iteratively until the Deep Q Network (a regular LSTM-RNN) learns the rules that were set at the beginning. After enough training steps, this process leads to some very consistent outputs which successfully adhere to the aforementioned rules and therefore feature substantial “musicality”, especially when compared to most works in this field.

The RL Tuner (Jaques et al., 2016) is clearly innovative and provides interesting approaches to automated music composition processes, but its full potential has arguably not yet been realised. Indeed, the RL Tuner offers ways to automatically adjust sequences of musical phrases generated by RNNs, but effectively only uses this method to perform general and relatively minor modifications on musical compositions. This is especially underwhelming since the rule-set used by the RL Tuner does not imbue the compositions with any discernible style or long-term structure, which again is a bottleneck in the area of artificial music composition.

In this article, we build upon the ideas and process developed by Magenta in the RL Tuner project and demonstrate how such an approach can be further developed and applied to a more intricate context in an attempt to solve the issue of lack of style and structure in automated music compositions. We present the results of extensive experimentation to provide a comprehensive perspective on artificially composing music that abides by particular principles in terms of rhythm, melody and form. In this process, we focus on the development of a novel rule-set (set of music theory rewards) derived from a musicological analysis of a particular musical genre that includes information about style as well as long-term structure.

Whereas previous work in automated music generation has investigated a variety of musical genres such as pop (Chu, Urtasun & Fidler, 2016) or classical music (Hadjeres, Pachet & Nielsen, 2017), as well as compilations of different pieces (e.g., piano-roll collections (Dong et al., 2018)), our focus is on folk music—a musical genre which is largely underrepresented in this field (having only been explored explicitly in a small number of works (Sturm et al., 2016)). In particular, we will focus on the Galician Xota, a genre that consists of precise musical characteristics and form (or long-term structure) (Schubarth & Santamarina, 1984), that allow for the extraction of standardized composition rules which can be generalized to the musical genre.

The remaining sections of this article are structured in the following manner. “Galician Xota” introduces the Galician Xota, the musicological analysis of this genre, the set of music rewards based on this analysis, and the creation of the dataset used in this article. “Methodology” introduces the two core machine learning (ML) methods used for our experiments—the Melody RNN (Magenta, 2019b) and the RL Tuner (Magenta, 2019c)—and “Experimental Procedure” outlines our methodology and experiments. The results and analysis of our experiments are described in “Results and Analysis”, and in “Conclusion” we provide our conclusions and outlook.

Galician xota

In this section, we provide the musicological analysis of the Galician Xota, which will be used to develop a system of music rules/rewards pertaining to this musical genre (also known as a musical rhythm (Foxo, 2007) or dance (Vásquez, 2010)). Furthermore, we also describe the process of creating a representative dataset for this musical genre, which will be used in our experiments.

Musical analysis

The Galician Xota is the variant of a Spanish traditional dance, named Jota in the Spanish language, present in the region of Galicia. The origin of the Jota is not fully clear, as shown by diverse presented theories (Olmeda, 1992). As most of the dances introduced in Galicia, the Xota arrived from outside of the region (Diario Oficial de Galicia, 2018). Some of the hypotheses suggest that dances were brought by foreigners through the Cami n¨o de Santiago (i.e., the Saint James Way) or through seasonal exchanges of Galician reapers working in neighbouring Spanish regions. The Galician Xota differs from other Spanish variants in the fact that assumed characteristics typical of traditional Galician dances, in particular the Muiñeira (Diario Oficial de Galicia, 2018). From a socioeconomic point of view, after the end of the Spanish dictatorship in 1975, a modern revival movement of the Galician culture promoted an ever-increasing interest in folk music (Vásquez, 2010). This phenomenon led to systematic ethnomusicological research in which a vast array of folk songs previously transmitted only by oral tradition, including the Xota, were recorded, musically transcribed, and published for the first time; see e.g., works by Martínez Torner & Bal y Gay (1973) or Schubarth & Santamarina (1984). Note that this is not trivial, since without written sources the presented research would have not been possible.

The Galician Xota presents two contrasting musical sections: the volta and the punto. By analyzing the pieces in the dataset, we can observe that the most common volta and punto are those identified by Schubarth & Santamarina (1984) as type one, i.e., melodies with two verses in which the second verse is the repetition of the first. From now on we will refer to these verses as section A for the volta and section B for the punto. Although some pieces pertaining to this genre alternate frequently between sections A and B, the structure presented in most of the MIDIs contained in our dataset is A A B B A A. Considering that each verse is musicalised in eight measures (bars), each piece generally presents a total of 48 measures, occasionally followed by a small number of additional measures at the end of the composition.

The Galician Xota is also characterised by a fixed triple meter, typically 3/8 and 3/4 (Schubarth & Santamarina, 1984), in which A and B are musically contrasting. Section A presents rhythms based on short notes (rule XA11 ), which encourage fast movements from the dancers across different positions, while B is based on long-note rhythms (rule XB1), which promote the development of a more acrobatic choreography which does not contemplate positional changes (Albán Laxe, 2003). Our analysis also revealed that section A commonly starts with the pickup beat (rule XA3) and does not present rhythmic syncopation, whereas B typically starts at the beginning of the measure, is highly characterised by syncopated rhythms (Schubarth & Santamarina, 1984), and generally ends with two quarter note rests (rule XB3). Given that section B starts at the beginning of the measure and considering that the presented repertoire is meant for a wind instrument, these two rests allow the player to breathe before the repetition of B. Finally, our rhythmical analysis shows that both A and B end with a quarter note (rule X1), which relates to the implicit composition principle that rhythms which end with a whole note transmit a conclusive sense (Roe, 1823).

Our melodic analysis of the dataset revealed that most of the considered melodies are based on the tonalities of D and C (major and minor), as well as F major. For the major tonalities, the natural scale was considered, while for the minor ones the melodic scale was used, i.e., the leading-tone (the seventh grade of the scale) was elevated by a half-tone when leading to the tonic note (the first grade of the scale) but was natural in descendent movements (Stefan Kostka, Dorothy Payne & Byron Almén, 2017). This implies that, when composing in minor tonalities, an additional note (the leading-tone) was considered in addition to the seven notes of the original scale. Notice that we have generalized our conclusions by focusing on the most common tonalities (for a detailed evaluation cf. Schubarth & Santamarina (1984)). Additionally, we observed that both A and B typically follow the sub-tonic, i.e., the note which comes before the tonic in this tonality, with the tonic note (rule X3) and frequently end the section with the tonic (rule X2), which is typical when following traditional harmonic rules (Rimsky-Korsakov, Joseph & Nicholas, 2005).

The melodic contour, as frequently seen in folk music, is mostly undulated (Schubarth & Santamarina, 1984), i.e., after a skip (two consecutive notes in disjunct motion) the melody develops in contrary motion (opposite direction) to the skip (rule X4). Similarly, our analysis showed that after a melody developed in conjunct motion (diatonic scale) until a maximum of six notes, its melodic contour would change direction (rule X5). Both sections of the Galician Xota are also contrasting from a melodic point of view: while A was mostly diatonic, characterised by the use of intervals of 3 rd (rule XA2), B showed more variety within its melody, featuring broader intervals such as 4 ths, 5 ths, and 6 ths, as well as some repeated notes (rule XB2). Finally, our musical analysis also revealed that the last four measures of a verse (for both A and B) were commonly a progression of the first four, meaning that the latter might be seen as a repetition of the former, starting in a different pitch class with small variations (e.g., in melodic intervals and rhythm).

Galician Xota dataset

The dataset was created by collecting MIDI files from the repertoire of Galician Xota for bagpipe available from a variety of online sources (Folkoteca Gallega, 2019; Gaita Gallega, Repertorio de Gaita, 2019; Galician Music Repository, 2019; Partituras de Gaita Galega, Partituras en Do, 2019). The total number of MIDI files collected2 was 496, with an average duration of 87.76 seconds and an average of 860 events per file. Some of these files were polyphonic, i.e., they featured multiple simultaneous tracks. Since the RL Tuner only supports monophonic input, i.e., featuring only one track, we separated the polyphonic pieces into monophonic MIDI files which could be loaded as input for our model, keeping only the tracks of substantial length (100 or more notes). The final dataset comprises 712 monophonic MIDI files. These files were split into a training set (80%) and a validation set (20%) based on the size (i.e., Bytes) of the MIDI files (the list of instances included in each set can be found in the project repository).

The small size of our Xota database is a potential overfitting factor during training. This is especially problematic in this project because we require the trained RNN to have broad probability densities in order to be flexible enough to adapt to the new rules during the RL Tuner phase. This diversity in note probabilities is evidently proportional to the amount of different training compositions. Gathering more original MIDI files pertaining to this very particular genre would be a potential solution, but it would also be a challenging one. For this reason, we opted for a data augmentation technique used in other works (Chen & Miikkulainen, 2001). This process consisted of transposing the training pieces to 24 new tones by transposing each note in a given piece by the same number of semitones. This process does not affect the musical characteristics of the original piece (in terms of rhythm and melody) and generates a much larger training set. In this case, it multiplied the size of our training set by 25, amounting to a total of 14,100 pieces.

Methodology

In this section, we describe the technical details of the two Magenta ML methods which served as the basis for our experiments—Melody RNN and RL Tuner—as well as the rule-sets used in our study.

Melody RNN

The Melody RNN (Magenta, 2019b) is an LSTM-based model developed by Google Magenta to compose discrete artificial music, and it consists of three variants: the Basic RNN, the Lookback RNN, and the Attention RNN. After close analysis of the Melody RNN code available at Magenta (2019b), it became evident that the Attention and Lookback RNNs are incompatible with the RL Tuner due to the way that music is encoded. Specifically, these networks use the lookback encoding (Waite, 2016), which does not only represent pitches, rests, and holds as the Basic RNN encoding that we describe below, but also encodes more complex behaviours such as note repetition. For this reason, we decided to focus our experiments on the Basic RNN, which is fully compatible with the RL Tuner. This model is described in the following paragraphs.

The Basic RNN is the simplest variant of the Melody RNN—a traditional LSTM-RNN similar to the ones used in various other works in this field (Brunner et al., 2017; Chu, Urtasun & Fidler, 2016; Eck & Schmidhuber, 2002). This network receives a one-hot vector representing the first note of the composition as input, it calculates a state which is modelled in the LSTM cell, and based on this state and the input it predicts the next note in the same one-hot format. The next state is then calculated, and the following input is loaded. Through this process, the network learns the temporal association between notes in the compositions featured in the dataset. After the training is complete, the network is primed with a random note, and the next note in the composition is determined using the network prediction. This predicted note is then loaded as the input and the generation continues iteratively until the maximum compositional length is reached.

The standard encoding used by the Basic RNN (and the RL Tuner) considers music pieces as one-dimensional vectors which constitute sequences of events. Each event consists of 38 information tokens: integers 2–37 represent 36 possible pitches spanning three octaves, 0 represents a held note event (the continuation of a pitch or a pause), and one represents a note-off event (ceasing to play a note or beginning of a pause). As an example, if we consider the sixteenth note as the smallest unit of representation, a quarter note would be represented as [X,0,0,0] (where X≥2) and a quarter rest would be represented as [1,0,0,0]. A direct translation between MIDI and this encoding can be seen in Fig. 1. This format can only represent monophonic music, which means that the RL Tuner and the Basic RNN must be trained with and compose pieces with only one note/event per time-step (i.e., no two notes can be sounding simultaneously).

Figure 1 Translation process from MIDI format (displayed using MIDI Toolbox (Eerola & Toiviainen, 2004)) to Basic RNN encoding for a simple monophonic melody.

The octave A3–A4 is displayed, with one beat equivalent to one quarter note.

RL tuner

The RL Tuner (see Fig. 2) is a direct application of the Deep Q Learning algorithm developed by Google DeepMind in 2016 (Mnih et al., 2013). It consists of three LSTM networks: the Deep Q Network; the Target Q Network, since we are using Double Q Learning (van Hasselt, Guez & Silver, 2015); and the Reward RNN. To fully understand the role of these three networks, it is necessary to first contextualise them by defining the Markov Decision Process (MDP) (Sutton & Barto, 1998):

Figure 2 The reinforcement learning tuner architecture, where s, a and r represent state, action, and reward respectively (Jaques et al., 2016).

The state space S—The state consists of the LSTM states of the Q-network and the Reward RNN combined with the composition so far.

The action-space A—The action is the note that we will compose next, which can be one of 38 possible values.

The transition probability Pss′a (the probability of going from state s to s′ given action a)—This concept is not explicitly defined in our framework, but the transition is determined by running the note through the reward RNN and the Q network and observing their new states.

The discount factor γ—This value between 0 and 1 determines how future rewards are valued by the agent. It is set to 0.5 for our experiments.

The reward function Rss′a—The reward depends partially on the Reward RNN and partially on music theory rules that decide the reward based on the action and the previous composition.

The policy π—This is what we seek to learn (which note should be played for a specific state of the composition).

The training procedure for this architecture is as follows. A trained RNN checkpoint (obtained after training the Basic RNN) is loaded by all three of the RNNs, which means they are initially identical. As the training progresses, the Reward RNN remains fixed while the Deep Q Network is updated using the traditional Deep Q Learning algorithm (Mnih et al., 2013) guided by the MDP, and the Target Q Network is gradually updated to resemble the Deep Q Network. When the training is concluded, the Deep Q Network is used to compose as a traditional LSTM-RNN network.

As mentioned previously, part of the reward given for an action in this MDP is given by the programmer. One can decide to give a reward based on the action (the next note) and the previous composition. Essentially, any pattern can be theoretically programmed to be rewarded, but this does not guarantee any impact on the post-RL compositions. This music theory reward is then determined by what we will denote as a rule-set which determines what kind of behaviour should be rewarded/punished.

Rule-sets

We used three rule-sets for the music theory rewards in our work. These are described in Table 1 and summarised in Table 2.

Table 1 Description of every rule included in the original Magenta rule-set, as well as every rule included in our Xota rule-set.

Rules beginning with XA apply only to section A and rules beginning with XB apply only to section B.

Rule code	Rule name	Description	
Magenta	
M1	Stay in key	The network is negatively rewarded for composing notes which are outside the key of the composition (defined a priori). For our experiments, the key was set to C major for all training and generation procedures.	
M2	Start and end with the tonic note	The agent is rewarded for starting and ending the composition with the tonic note. The implementation of this rule is somewhat inadequate since it only rewards the agent for composing the tonic note in the first and last event of the composition. This means that the reward is not applied when the composition starts with a rest followed by the tonic note, for instance, which is arguably undesirable.	
M3	Avoid excessive repetition of the same pitch	This rule is applied in terms of a reward function which penalises the composition for writing more than eight repeated notes in a row with pauses and held notes in between, more than six repeated notes with pauses or held notes in between, or more than four repeated notes with no pauses or held notes in between.	
M4	Prefer harmonious intervals	This rule rewards the agent for composing harmonious intervals such as perfect fifths or major thirds in key and punishes intervals such as sevenths or eighths since they generally sound jarring.	
M5	Resolve large leaps in pitch	This rule is implemented through a function that detects the leap (an interval larger or equal to a perfect fifth) and its direction, and then rewards the composition if it detects a similar leap in the opposite direction within the next six notes. This is an extremely well-realised rule that successfully promotes tonal consistency, which is one of the most difficult tasks in music composition.	
M6	Avoid repeating pitch extrema	The agent is punished for using the highest and lowest pitches in the composition more than once. The reward is only attributed at the end of the composition.	
M7	Avoid high autocorrelation with previous notes	This rule is applied by calculating the correlation between the current composition and the previous composition with a lag of 1, 2, and 3 events. If any of these are relatively high, a negative reward is emitted. The correlation is calculated using the formula defined in (Magenta, 2019c).	
M8	Play motifs	This rule aims to incentivise motifs by rewarding the composition of musical chunks which have the general format of a motif: a sequence of eight notes/events with three or more unique pitches.	
M9	Play repeated motifs	This rule is enforced through a reward function that checks whether there was a motif in the previous eight notes (including the action) and if so, checks if it was present in its exact form throughout the whole previous composition. This rule is substantially less effective than the rest of the rule-set.	
Xota	
X1	Sections end with a quarter note	Every section is rewarded for ending with a quarter note. Specifically, using the standard encoding mentioned in “Methodology”, sections are rewarded for ending in the following manner: [...,X,0,0,0], where X can be any note (2–37).	
X2	Sections end with the tonic	Every section should be rewarded if the last note being played (excluding note-on or note-off events) is the tonic note. This implies that any of the following compositions should receive this reward if we are at the end of a section: [...,4,5,T]; [...,T,0,0]; [...,T,1,0,1,0], where T is the tonic note for this tonality.	
X3	Tonic always follows sub-tonics	Any sub-tonic note in the composition which is directly followed (excluding note-on or note-off events) by the tonic should entail a reward. This means that any of these examples should be rewarded: […, ST, T, …], [...,ST,0,1,0,T], where ST is the sub-tonic and T is the tonic.	
X4	After a pitch interval larger than a 3rd, play the opposite note	After an interval of at least a 4th in a specific direction, the agent should be rewarded for composing the note directly above or below this last note (the one that opposes the melodic direction of the leap). In practice, this is achieved by monitoring the interval between the last two notes and providing a large reward for composing the opposing note in direct sequence to this event.	
X5	Reverse melodic direction	After three or more notes in the same direction, the agent should be rewarded for composing a new note in the opposite direction. This rule is meant to encourage the undulated melodic contour which is seen frequently in the Xota, as well as many other folk genres. In practice, the reward that is given for composing an opposite note is proportional to the number of previous notes in the same direction.	
XA1	Compose with fast notes	In section A, the agent should be rewarded for composing mostly in sixteenth notes, with some eighth notes and a few quarter notes also allowed. This is done by attributing different rewards to each of these note lengths in order to achieve the right balance, which in section A makes for a composition with faster notes.	
XA2	Compose thirds	In section A, the agent should be rewarded for composing with third intervals. The value of this reward was carefully tuned to that thirds are frequent while still occasionally allowing for other melodic intervals.	
XA3	Start with a quarter rest	Starting section A with a quarter rest should entail a reward. Specifically, this means that only this behaviour is rewarded, at the beginning of section A: [0,0,0,0,...].	
XB1	Compose with slow notes	In section B, the agent should be rewarded for composing few sixteenth notes, many eighth notes, and some quarter notes.	
XB2	Compose broad intervals	In section B, the agent should be rewarded for composing mostly in fourths, fifths, sixths, and sevenths. Once again, the value of this reward was tuned in order to ensure a variety of intervals would be featured in section B.	
XB3	End with two quarter rests	Ending section B with two quarter rests (or a half rest) should entail a reward. This rule was implemented to ensure that the last two thirds of the final bar of section B would be silent.	

Table 2 Description of the three different rule-sets explored in this work, showing which rules are contained in which rule-set.

(A) and (B) represent the rules that are specific for each section (A or B).

Rule	Rule-set	
#	Name	Performance Measure	Magenta	Xota	Magenta+ Xota	
M1	Stay in key	Percentage of notes in key	✓		✓	
M2	Start and end with the tonic note	Percentage of compositions starting and ending with the tonic note	✓			
M3	Avoid excessive repetition of the same pitch	Percentage of non-repeated notes	✓		✓	
M4	Prefer harmonious intervals	Percentage of intervals which are perfect fifths or major thirds in key	✓		✓	
M5	Resolve large leaps in pitch	Percentage of leaps which are resolved	✓		✓	
M6	Avoid repeating pitch extrema	Percentage of compositions featuring unique extrema	✓			
M7	Avoid high auto-correlation with previous notes	Inverse of the auto-correlation with the previous three notes (100 subtracted by the auto-correlation in percentage)	✓		✓	
M8	Play motifs	Percentage of notes in motifs	✓		✓	
M9	Play repeated motifs	Percentage of notes in repeated motifs	✓		✓	
X1	Sections end with a quarter note	Percentage of compositions ending with a quarter note		✓	✓	
X2	Sections end with the tonic	Percentage of compositions ending with the tonic note		✓	✓	
X3	Tonic always follows sub-tonics	Percentage of sub-tonic notes which are followed by the tonic		✓	✓	
X4	After a pitch interval larger than a 3rd, play the opposite note	Percentage of leaps which are followed by the opposite note		✓	✓	
X5	Reverse melodic direction	Percentage of intervals which reverse direction		✓	✓	
XA1	Compose with fast notes	Percentage of notes which are 1/8 or 1/16 in length		✓(A)	✓(A)	
XA2	Compose thirds	Percentage of intervals which are thirds		✓(A)	✓(A)	
XA3	Start with a quarter rest	Percentage of compositions starting with a quarter rest		✓(A)	✓(A)	
XB1	Compose with slow notes	Percentage of notes which are 1/4 or 1/8 in length		✓(B)	✓(B)	
XB2	Compose broad intervals	Percentage of intervals which are larger or equal to a perfect fourth in pitch		✓(B)	✓(B)	
XB3	End with two quarter rests	Percentage of compositions which end with one or two quarter rests		✓(B)	✓(B)	

Magenta rule-set

The intention of Magenta’s original rule-set (Jaques et al., 2016) is to tune compositions in a non-specific style. The motivation for this comes from the fact that music pieces are sequences that typically follow a set of rules that apply to most genres and styles, but RNNs generally fail to model and apply these rules consistently. This rule-set introduces a set of general, well-established rules that new compositions should adhere to, and none of them address issues related to form or long-term structure. Mostly, it focuses on correcting the frequent issues of pre-trained RNN compositions (e.g., extreme repetition or auto-correlation) and endowing the network with the capacity to compose music with characteristics that tend to be common in most western music (e.g., starting the composition with the tonic note, remaining in the same key throughout the composition). A description of all the rules from this rule-set is included in Table 1 (rule codes M1 to M9).

Xota rule-set

This new rule-set was created based on the musicological analysis detailed in “Methodology”, and it aims to describe in computational terms the stylistic and structural rules that characterise the Galician Xota. Since this rule-set includes instructions related to musical form and long-term structure, some of the rules apply only to specific sections (A or B), whereas others apply to the whole composition. The structure of the composition regarding sections and events is illustrated in Fig. 3. A description of all the rules from this rule-set is included in Table 1—rule codes X1 to X5 apply to all sections, XA1 to XA3 apply only to sections of type A, and XB1 to XB3 apply only to sections of type B.

Figure 3 This scheme illustrates the format used for the RL Tuner compositions.

Each event represents a sixteenth note.

Magenta+Xota rule-set

Since the Xota rule-set is specifically tailored for one genre, it may be possible that it lacks general music rules which virtually any musical genre typically adheres to (e.g., avoiding excessive repetition). For this reason, we decided to combine the Magenta and the Xota rule-sets and form a third rule-set containing every rule in both sets with the exceptions of rules 2 and 6 of the Magenta rule-set because they are not compatible with the rules devised for the Galician Xota.

Experimental procedure

In this section, we describe the methodological and technical aspects of the experiments performed in the context of this work.

Experiments

Regarding the Basic RNN training we conducted experiments in three different scenarios: M: Magenta Basic RNN—This configuration uses the pre-trained Magenta checkpoint (Magenta, 2019b), whereby the Basic RNN was trained with a large dataset composed of thousands of different MIDI files pertaining to various genres and styles (the Magenta dataset).

X: Xota Basic RNN—For this configuration, the Basic RNN is trained with the Xota dataset mentioned in “Galician Xota”.

MX: Magenta+Xota Basic RNN—For this configuration, the Magenta Basic RNN is re-trained with the Xota dataset, as we will describe below.

After training, each of the Basic RNN checkpoints was loaded into the respective RL Tuner, and we experimented with different conditions to fine-tune the models: 0. No rule-set (baseline)—RL training is not performed

1. Magenta rule-set

2. Xota rule-set

3. Magenta+Xota rule-set

In total, the combinations of RNN checkpoints and rule-sets which were tested in this work are enumerated in Table 3, which attributes a numerical identifier to each configuration for future reference. During training and generation, the compositions were generated section by section, so as to not increase the state space exponentially.

Table 3 Summary of experiments: Magenta, Xota, and the combination of both.

	Magenta basic RNN	Xota basic RNN	Magenta+Xota basic RNN	
Pre-RL	M.0	X.0	MX.0	
Magenta rule-set	M.1	X.1	MX.1	
Xota rule-set	M.2	X.2	MX.2	
Magenta+Xota rule-set	M.3	X.3	MX.3	

Parameters

As previously mentioned the Basic RNN is a traditional LSTM-RNN. In our experiments, we trained a Basic RNN with the Xota dataset using a network consisting of two hidden layers with 512 units each with an input and output size of 38 (referring to the 38 possible events). The dropout was kept at 0.5 to prevent overfitting, and the gradient clipping norm was set to five to avoid numerical instability due to exploding gradients. The learning rate was optimised by applying the Adam optimizer (Kingma & Ba, 2014), using an initial value of 0.001 ( β1=0.9, β2=0.99). During training, the data was split into batches of size 128 (which were then randomly selected).

Regarding the training of the RL Tuner, we applied the Q-learning algorithm using an ϵ-greedy exploration mode. The layers of the Deep Q Network, Target Q Network and Reward RNN were equivalent to the ones mentioned above, to ensure compatibility. The number of training steps was set to 1,000,000 for all the experiments, whereas the number of exploration steps was set to 500,000. The reward scaler, which re-scales the music theory reward, was set to 2.0 given the importance of the music theory reward for this work. The discount factor was kept at 0.5 and the mini-batch size for the experience replay (Mnih et al., 2013) was set to 32 for all experiments. The Target Q Network update rate was kept at 0.01 (as in the original code (Magenta, 2019c)). In addition, the number of events per piece was 96 for all experiments, since we were composing each section individually.

Combining magenta and xota

Given that our dataset is specific to one genre, it is not as representative of general aspects of music structure as many of the corpora typically used for music generation. Therefore, in order to consider more general music rules, we decided to combine the Xota dataset with the dataset used by the Magenta models. We did so by re-training the pre-trained Magenta model (trained on the Magenta corpus (Jaques et al., 2016)) with the Xota dataset. We refer to this configuration as the Magenta+Xota RNN. The parameters used for this training procedure are the same as the ones mentioned above for the Xota Basic RNN.

Managing overfitting

Overfitting is a recurring issue when training RNNs, and a major concern in this work given that the models may recreate chunks of the training data (existent music pieces) rather than compose original music in that style. This problem is particularly important in our work given that we aim to change the behaviour of the RNN after training it with the Xota dataset, which means that the RNN must be flexible enough to adopt new behaviours. If we fail to achieve this flexibility, the RNN will become overconfident and will not adhere to the music theory rules we set out. We illustrate this issue in Fig. 4. In the context of this project, we faced this issue in preliminary experiments when we attempted to train the Basic RNN with the Xota dataset using 20,000 training steps as suggested in (Magenta, 2019b)—an excess of training steps was causing the model to overfit. To solve this issue, we monitored the validation accuracy of our models during 5,000 training steps (see Figs. 5 and 6), and kept the checkpoint that achieved the peak validation accuracy. The Xota Basic RNN reached its maximum validation accuracy of 78.27% at training step 1,351, while the Magenta+Xota Basic RNN reached its maximum validation accuracy of 79.38% at training step 301 (much earlier due to its previous training with the Magenta dataset).

Figure 4 The issue of overfitting.

The first network (A), which has been trained for almost 5,000 steps, is very confident about the notes due to its high accuracy and low perplexity (this makes it less flexbile, which will make it more difficult to learn new rules). The second network (B), which has been trained for fewer steps, is much more flexible and has various different possible notes at every time-step (this means it will adapt more easily to the music theory rules).

Figure 5 Training accuracy and validation accuracy during Xota Basic RNN training procedure.

It can easily be noted that the validation accuracy reaches its peak between 1,000 and 1,500 training steps, while the training accuracy keeps increasing indefinitely.

Figure 6 Training accuracy and validation accuracy during Magenta+Xota Basic RNN training procedure.

The validation accuracy clearly peaks during the first 500 training steps and then stagnates, while the training accuracy keeps increasing.

Results and analysis

For each experiment shown in Table 3, we generated 1,000 compositions by priming the RNN with a single random note and composing a total of 96 events per composition (one full section). Our description and analysis of the results will focus on two aspects. First, we evaluate the performance of the three basic RNNs (Magenta, Xota and Magenta+Xota) before applying the RL Tuner in order to understand how well the models were able to extract relevant information pertaining to each rule during training. Then, we evaluate the impact of the RL Tuner training procedure on the models’ performance, applying each of the rule-sets we defined. The performance on each rule (hereinafter, scores) was computed using the performance measures described in Table 2 and averaged across the 1,000 generated compositions.

Scores referring to rules applied to both sections of the compositions were measured for sections A and B separately and subsequently averaged. The Pre-RL (_.0) scores, the difference between Post-RL and Pre-RL scores (i.e., M.1 - M.0, M.2 - M.0, etc.) and the Post-RL scores (_.1, _.2 and _.3) are shown in Figs. 7–9, respectively. Scores are measured in percentage (as mentioned in Table 2) and range from 0 (low performance) to 100 (high performance). The difference between Post-RL and Pre-RL scores provides a simple overview of the effect of the RL Tuner training on the outputs of these networks—positive scores represent improvements in a specific rule after using the RL Tuner, whereas negative scores represent a decline in performance. In all three figures, the scores for each rule were rounded to the nearest integer.

Figure 7 The Pre-RL scores of each rule in the Magenta+Xota rule-set during our experiments with every RNN/rule-set configuration.

The numerical identifiers for each rule and configuration are described in Tables 2 and 3, respectively.

Figure 8 The difference between the Post-RL and Pre-RL scores of each rule in the Magenta+Xota rule-set during our experiments with every RNN/rule-set configuration.

The numerical identifiers for each rule and configuration are described in Tables 2 and 3, respectively.

Figure 9 The post-RL scores of each rule in the Magenta+Xota rule-set during our experiments with every RNN/rule-set configuration.

The numerical identifiers for each rule and configuration are described in Tables 2 and 3 respectively.

Pre-RL

We start by looking at the performance of the RNNs before any RL tuning. The average scores for the Magenta rule-set ( MAvg) show that X.0 achieved the best results— MAvg=64%—closely followed by MX.0— MAvg=63%. Interestingly, both these models outperformed M.0 on several rules, suggesting that even a model trained with very few music pieces (X.0) can achieve better rule scores than the original Magenta RNN. This may indicate that the Xota dataset is a better representative of the Magenta rules than the dataset collected by Magenta. This is plausible given that the Magenta dataset was not designed to align with the Magenta rule-set. We hypothesize that this behaviour may be due to the nature of these two datasets. While the Xota dataset is composed of pieces from a single traditional folk music genre, which, in our experience, seems to follow the Magenta rules to some extent, the Magenta dataset contains thousands of MIDIs with no specific genre. These pieces may be taken from genres that do not adhere to the traditional rules designed by Magenta as accurately as the pieces in the Xota dataset, which would explain the low scores achieved by the Magenta RNN. We highlight, however, that this is merely a hypothesis, as we do not have access to the Magenta dataset and therefore cannot make any definitive claims about its contents, and how they may relate to the Magenta RNN’s performance. For most rules in this rule-set, the performance of all model configurations is similar, except for the performance on rule M8 which seems to be responsible for the best results obtained with X.0. Indeed, the compositions produced by X.0 exhibit a larger amount of motifs—on average, 79% of the notes are integrated into a motif, compared to 23% for M.0 and 58% for MX.0.

Regarding the Xota rule-set, the average performance ( XAvg) is much lower when compared to the Magenta rule-set ( MAvg) and is similar for all configurations, with a slightly better performance obtained with X.0 and MX.0. It should also be noted that all models perform poorly on some of the rules (e.g., M4, M9, X1, X4, X5, XA3, XB3), indicating that the training procedure was not able to extract relevant information from the compositions to generate new music following these principles. Conversely, all models perform very well for several rules and achieve nearly maximal scores in some cases (e.g., M1, M3, M5, M7), and consequently the RL phase will have very little margin for improvement.

The impact of the RL Tuner on the models’ performance

In order to interpret the effect of the RL phase on the models’ performance we will initially focus on Fig. 8, which displays the difference between the Post-RL and Pre-RL scores obtained for each rule. In this figure, improvements in the scores on each particular rule and on a specific experiment are marked in green, whereas scores that decreased are marked in red (the magnitude of the change is proportional to the circle radius). Generally, all models led to improvements in the Magenta rules scores, with model M.1 (Magenta RNN fine-tuned with the Magenta rule-set) showing the largest improvements ( MAvg(Post_RL)−MAvg(Pre_RL)=13%). Whereas the performance for most Magenta rules improved (even if slightly), we can also observe some degradation, especially for rules M5, M7, and M8 (when using the X._ basic configurations). Nonetheless, these rules already had high scores in the Pre-RL phase (see Fig. 7) and, in some cases, degradation did not have a meaningful impact on the absolute performance. It is interesting to observe that although M.0 showed a poor performance for rule M8, the RL tuning largely improved this rule in experiments M.1 and M.3. The opposite happened in experiments X.1, X.2 and X.3—whereas model X.0 had the best performance on this rule in the Pre-RL phase, RL tuning led to a large decrease in this score.

Regarding the Xota rule-set, experiments M.2, M.3, and MX.3 show the largest improvements in rule scores ( XAvg(Post_RL)−XAvg(Pre_RL)=23% for M.2 and MX.3, and XAvg(Post_RL)−XAvg(Pre_RL)=30% for M.3), effectively doubling the scores when compared to the Pre-RL phase. Considering our aim to produce compositions in the Xota style, it is worth noticing that both M.3 and MX.3 were fine-tuned with the combined rules from the Magenta and Xota rule-sets, which indicates that the rule-set created in this work led to compositions that better adhere to the Xota style than the ones produced by the original Magenta rule-set. This is confirmed by the fact that the smallest improvements in the Xota rules were obtained with the Magenta rule-set alone.

It is also important to notice that the fine-tuning of the models with the Xota rule-set did not perform as well as the combinations of the Magenta+Xota rules, suggesting that more general music rules not specific to the music style are important for developing style-specific models. Finally, these results also suggest that whereas X.0 led to the best performance in the Pre-RL phase for both rule-sets, the RL tuning had a strong negative impact on its performance regarding the Magenta rules M7 and M8. This strong Pre-RL performance is explained by the fact that the Xota RNN (X.0) composes almost exclusively sixteenth notes (shown by its performance in rules XA1 and XB1), which are represented by one event in the Basic RNN encoding and are considered fast notes. This leads us to rules M7 and M8, which measure correlation and percentage of notes in motifs. Given how these rules enforce and measure these behaviours, mentioned in Tables 1 and 2, it is evident that their performance will improve when the density of pitches is higher i.e., when the notes are faster. This is why X.0 has such a high performance for these rules, whereas X.1, X.2, and X.3 have comparatively low performances since they are tuned to compose slower notes (especially in section B).

Post-RL

Looking now at the absolute values obtained in the Post-RL phase, we can see that MX.1 was the configuration that led to the best performance on the Magenta rule-set ( MAvg=68%), closely followed by M.1 ( MAvg=65%). Compared to the Pre-RL phase, this indicates a performance improvement of roughly 5% for MX.1 (compared to MX.0) and 13% for M.1 (compared to M.0). If we consider the performance of X.0 as a baseline (the best model in the Pre-RL phase— MAvg=64%), then, the improvements are of 4% for MX.1 and 1% for M.1. This indicates that RL tuning had a very small impact on the performance scores of the Magenta rule-set if we started with a model trained on either the Xota database (X._) or the Magenta models retrained with the Xota pieces (MX._). This is because, before the RL Tuner, these RNNs (X.0 and MX.0) generally had higher scores than the original Magenta RNN (M.0). This means that they were less affected by the training procedure using the Magenta rule-set, since they did not have as much need for improvement in this regard.

Regarding the Xota rule-set, M.3 led to the best performance ( XAvg=46%), followed by MX.3 ( XAvg=40%). Compared to the Pre-RL phase, this indicates a performance improvement of about 30% for M.3 (compared to M.0) and 23% for MX.3 (compared to MX.0). These results show that RL tuning had a strong impact on the performance scores of the Xota rule-set, more than doubling the performance. Therefore, the combined rule-set (_.3) was very effective at enforcing the Xota rules, and was even more effective than the Xota rule-set alone (_.2). Clearly, combining general and genre-specific music rules was more effective than a single set of specific rules when attempting to emulate this musical genre. Additionally, it is worth highlighting that, although configurations M.3 and MX.3 both did well, M.3 was superior for both rule-sets (Magenta and Xota). This suggests that RNNs trained only with a larger, non-genre-specifc dataset (in this case, the Magenta dataset) led to better results when trained to adhere to a new rule-set.

The nature of the rules on the RL tuning

Considering the Post-RL results, it is important to highlight some aspects regarding the design of rules for the RL Tuner. The first of these is that rules that are computed more frequently also have a more frequent impact on the rewards, which means that some rules may not be as effectively acquired by the model since they are not affecting the reward value as often. An illustrative example is the difference between rules X1 and XB3—whereas the first rewards any quarter note at the end of the composition and achieved an improvement of 16% during the RL phase for configuration M.3, the second rewards only quarter rests (a more specific and therefore less frequent behaviour) and led to low improvements (below 10%) for almost every configuration. Another important factor is the position in time of the rewarded behaviours—rewarding early behaviours is much more effective since the state space increases with time (at t=1, there are 38 possible states, at t=2 there are 382, and so on). This can be seen if we contrast rule XA3 with XB3—rule XA3 (rewarding early rests) had improvements greater than 70% for four configurations, whilst rule XB3 (rewarding late rests) achieved low scores in general.

Another important conclusion derived from our results is that complex rules are generally more difficult to learn for the Deep Q Network. This is because these rules typically apply after a complex chain of events, which can make it difficult for the network to learn which action led to the reward. In addition, the application of this reward can be rather infrequent, which means there are fewer examples of the desired behaviour in the training experiences. Rules M5, M7, X4, and X5, for instance, are more complex than most of the other rules featured in our rule-sets, since they either require pre-conditions like a leap in the composition, or reward complex behaviour such as a change in the direction of the pitch contour. Due to this fact, these rules all scored improvements below or equal to 10% for every configuration (see Fig. 8).

Subjective evaluation

In order to evaluate the musicality, structure, and characterization of the generated samples from a perceptual point of view, we perform a subjective evaluation of the generated samples. The subjective evaluation was carried out by a musicologist trained in Galician Folk music through a listening test. In a questionnaire, the coherency of five randomly selected samples generated by each method, i.e., a total of 60 musical pieces were assessed with respect to the typical characteristics of the Galician Xota. Four musical parameters were considered as evaluation criteria: (i) melody, i.e., the general coherency of the melodic contour; (ii) rhythm, i.e., the rhythmic discourse; (iii) structure, i.e., to which extent the piece was temporally cohesive, perceptually well-structured and naturally followed the expected A A B B A A form; and (iv) formal rhythmic-melodic characterization, i.e., whether the rhythms and melodies used in the different parts were suitable to dance volta and punto. Each parameter was rated on a scale from 1 to 5, where 1 is not coherent at all, and five is very coherent. Finally, an overall score was obtained for each model by averaging the four criteria.

The results of the subjective evaluation are summarised in Table 4. The Magenta rule-set yielded the worst-performing model amongst the three proposed rule-sets from a perceptual point of view. Its compositions featured rhythms and melodies that sounded particularly unnatural, as reflected in the low melody and rhythm scores, but still managed to yield some improvements in terms of structure. Conversely, the models trained without any RL tuning (Pre-RL) were perceptually superior or equivalent in terms of general musicality, i.e., the melody and rhythm were perceived as more coherent. However, as expected, the structure and stylistic characterization of the pieces generated without rules did not correspond to the target genre, leading to very low scores in the latter two metrics. The models tuned with both rule-sets performed somewhat in between: rhythm and structure were generally reasonable, but the melody often sounded unnatural due to unexpected intervals which disrupted the expected melodic contour. As with the Magenta rule-set, the structure is noticeably improved after RL tuning.

Table 4 Summary of the subjective evaluation.

Mean scores across the five assessed samples generated by each method are given on a one (lower coherence) to five (higher coherence) scale. Mean results for each evaluation criterion (melody, rhythm, structure, and characterization) as well as an overall score (obtained by averaging the four criteria) are displayed. The best results are highlighted in bold. In the upper row, “Charact.” stands for characterization.

	Melody	Rhythm	Structure	Charact.	Overall	
Magenta Basic RNN (M)	
Pre-RL (M.0)	1.6	1.8	1.0	1.2	1.40	
Magenta rule-set (M.1)	1.2	1.2	2.4	1.0	1.45	
Xota rule-set (M.2)	2.0	2.2	2.8	1.2	2.05	
Magenta+Xota rule-set (M.3)	2.2	2.4	2.6	1.2	2.10	
Xota Basic RNN (X)	
Pre-RL (X.0)	1.4	1.2	1.0	1.0	1.15	
Magenta rule-set (X.1)	1.4	1.2	2.6	1.0	1.55	
Xota rule-set (X.2)	1.8	3.0	3.4	1.4	2.40	
Magenta+Xota rule-set (X.3)	1.6	1.8	2.6	1.2	1.80	
Magenta+Xota Basic RNN (MX)	
Pre-RL (MX.0)	2.8	2.8	1.0	1.0	1.90	
Magenta rule-set (MX.1)	1.4	1.4	2.6	1.4	1.70	
Xota rule-set (MX.2)	3.2	3.8	3.4	2.4	3.20	
Magenta+Xota rule-set (MX.3)	1.6	2.0	2.8	1.4	1.95	

Finally, according to this subjective evaluation, the models that performed best were the ones tuned with the Xota rule-set. Concerning the melody, although some unexpected (usually dissonant) intervals took place eventually, interrupting the natural musical flow, this was the exception rather than the rule. Similarly, the structure was mostly correct and the rhythm was generally quite coherent for the songs produced with the new rule-set. Nevertheless, abrupt rhythmic closures along with inconclusive cadences led to poor characterization, decreasing the overall score. In general, for all models and rule-sets, a coherent harmonic discourse was found to be missing from all generated samples, i.e., the pieces did not appear to have a clear tonal center. Remarkably, despite not containing any sections-specific rules, the Magenta rule-set provided moderate improvements in structure, which is likely due to rules such as M2, M5, and M9, which enforce temporal cohesion within each section.

While the outlined rule-sets roughly follow the same trends for all three Basic RNNs, there are also some noticeable differences between them. In particular, the Magenta+Xota Basic RNN (MX) achieves the highest scores for most rule-sets, indicating that fine-tuning the Magenta Basic RNN on the Xota dataset yields the best perceptual results, including before any RL tuning (MX.0). Interestingly, the Xota Basic RNN benefited greatly from RL tuning, but yielded the lowest overall perceptual score of 1.15 for Pre-RL (X.0), in contrast to the high scores achieved in Fig. 7. This suggests that the model trained on the relatively small Xota dataset produced compositions that followed the rules adequately but sounded unnatural and lacked musicality, which could be related to the lack of training data.

Overall, we draw three major conclusions from this subjective evaluation. The first is that the Xota rule-set improves the perceptual musicality, structure, and style of the generated compositions, as demonstrated by high scores across four criteria. Secondly, we find that despite yielding some improvements in structure and characterization, the Magenta rule-set yields compositions that are perceptually similar, or even worse than the Pre-RL model within the context of the Galician Xota. We hypothesize that despite being well-motivated in musical terms, the Magenta rules are enforced too strictly, potentially causing the model to forget the melodic and rhythmic patterns that were learned during training. Thirdly, we find that the Magenta+Xotas Basic RNN yielded the best compositions from a perceptual point of view. We conclude that, even if no RL tuning is involved, it is better to fine-tune a pre-trained model (trained on a large dataset of MIDIs) on our genre-specific dataset, rather than train one from scratch.

From a general point of view, we found that the best-performing models (tuned with the Xota rule-set) could produce compositions that eventually contained some high-level patterns typical of the investigated repertoire. In particular, we observed the use of repeated notes at the beginning of the musical phrases, which is reminiscent of real pieces such as the Xota do Barrio do Ceo. Despite this, we also found that even the pieces by the best-performing models were lacking a repetitive component in terms of musical motifs. This could be improved in future models by designing a more effective rule to encourage consistent motifs that are repeated at multiple points in the composition. Indeed, after the subjective evaluation, we conclude that simple progressions, i.e., repetitions of the first four measures of a verse starting in a new tonic (see Section “Galician Xota”), are a feature that appears to be crucial for the characterization of the Galician Xota.

Conclusion

In this work, we extended the RL Tuner architecture to compose in a specific musical style based on the Galician Xota, a folk music genre. To achieve this goal, we experimented with multiple datasets and, of course, an original rule-set meant to emulate this new style. This involved two distinct training procedures: training an RNN, and also training a Deep Q Network. These training procedures and the results obtained from the various experiments performed were described in detail. We finished by analyzing these results, which show a promising future for this architecture, and explained why some configurations of this model were more successful than others.

By using broad datasets (Magenta and Magenta+Xota) and combining the rule-set from the original RL Tuner with our new Xota rules, we achieved training configurations that reliably followed most of the behaviours we had observed in the dataset, such M.3 and MX.3. Furthermore, by conducting a subjective evaluation of the generated pieces (performed by an expert musicologist), we find that applying the RL Tuner with the Xota rule-set substantially improves the musicality, structure, and characterization of the compositions within the context of this genre. This is an important achievement since it shows that the RL Tuner can indeed be used not only for tuning general aspects of the generated compositions but also to provide them with style and structure, which are qualities that most artificial composers are not able to produce. On the other hand, we find that despite accurately following the rules set out in the Magenta+Xotas rule-set, models trained with the Magenta/Magenta+Xota rule-sets yield inferior performance in the subjective evaluation. We hypothesize that this behaviour is due to the Magenta rule-set being too restrictive, taking away from the musicality of the original (Pre-RL) composer.

In addition, by experimenting with many different parameters and configurations, we can describe several important factors for successful results when using this model, which are: the frequency of behaviour, the distance between action and reward, and the complexity of behaviour. This entails that to be successful with the RL Tuner, it is necessary not only to tune its hyperparameters but also to design rules which are optimal with respect to these factors as we have described in the previous sections. These conclusions are motivated by experimental results and will surely be valuable for any further research on this topic.

This work has also highlighted some of the limitations of our approach, which are mainly related to the effectiveness of enforcing specific music theory rules by performing Deep Q learning. Some rules struggled to have an impact on the final compositions even after training our Deep Q Network with a large number of training steps. This empirical insight clearly shows that designing a rule-set that is effective in the context of this model can be very challenging and typically must involve some trial and error. Namely, rules which encouraged infrequent and very specific behaviours were shown to be less impactful on the generated compositions since they were very rarely applied during training experiences.

Regarding future work, there are two extensions to our work that we would like to highlight. The first of these is related to Inverse Reinforcement Learning (IRL). Tuning the values for the rewards was the most time-consuming part of this project. Nevertheless, this is a very important process which is critical for generating compositions that adequately adhere to the rules which have been set. To prevent this laborious process, it would be productive to experiment using modern IRL techniques to automate and perfect this process. This would allow us to find the optimal reward values for each rule without requiring manual trial and error, which would improve performance and reduce time constraints in future projects. The other extension relates to rule-sets not needing to be exclusively related to music structure. Currently, we are experimenting with the concept of rule-sets for composition of musical forms that can convey emotional meaning (Coutinho & Dibben, 2013; Scherer & Coutinho, 2013). Based on the success of our Xota rule-set, we believe it would be viable to elaborate an emotional rule-set which could steer RNN-based composers toward specific emotional profiles.

Supplemental Information

Supplemental Information 1 Galician Xota Composer Code.

The code we used for our experiments and a guide on how to reproduce our results and visualizations.

Click here for additional data file.

Supplemental Information 2 Galician Xota Dataset.

The data, code and instructions on how we gathered and preprocessed its contents.

Click here for additional data file.

Supplemental Information 3 The results referenced in our article (network checkpoints, statistics, generated MIDI files, among others).

Click here for additional data file.

We thank Georgios Rizos for his valuable advice and insightful discussions.

Additional Information and Declarations

Competing Interests

Author Contributions

Data Availability

1 These rule names will be used later in the “Methodology” section to refer to individual rules derived from the musicological analysis and to set our rewards rule-set.

2 Some of the files gathered were in Guitar Pro 5 (GP5) format and were converted to MIDI.

Björn Schuller is an Academic Editor for PeerJ.

Rodrigo Mira conceived and designed the experiments, performed the experiments, analyzed the data, performed the computation work, prepared figures and/or tables, authored or reviewed drafts of the article, and approved the final draft.

Eduardo Coutinho conceived and designed the experiments, analyzed the data, prepared figures and/or tables, authored or reviewed drafts of the article, and approved the final draft.

Emilia Parada-Cabaleiro conceived and designed the experiments, performed the experiments, analyzed the data, prepared figures and/or tables, authored or reviewed drafts of the article, and approved the final draft.

Björn W. Schuller analyzed the data, authored or reviewed drafts of the article, and approved the final draft.

The following information was supplied regarding data availability:

The project repository containing code for all our experiments as well as the full dataset collected for our project (with instructions on how to collect and preprocess it), are available in the Supplemental File and on GitHub: https://github.com/miraodasilva/GalicianXotaComposer.

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
