# Peer review of "Automated composition of Galician Xota—tuning RNN-based composers for specific musical styles using deep Q-learning"

_PeerJ Computer Science, doi:10.7717/peerj-cs.1356_

## Round 0.1 · original submission · Major Revisions

The judgments of both reviewers are in fact quite similar. Both appreciate the purpose of the paper and its contribution to the state of knowledge. However, Reviewer 1 correctly identifies an issue with the experimental design that should be addressed. For that reason my recommendation must be "Major revisions".

Both reviewers also identify gaps in contextualising the work, and addressing external validity e.g. by subjective evaluation.

Please do attend to all of the feedback they provide, as it seems to me well-considered.

·

Excellent Review

This review has been rated excellent by staff (in the top 15% of reviews)
EDITOR COMMENT
Great review with precise and concise detail. Thank you!

Basic reporting

1.1 Language

The article is well written and the problem, methodology used, and findings are well explained. However, there are some ambiguities & typos and I would suggest proofreading being made to meet the criteria of clear, unambiguous, professional English language. In the additional comments, I included some of the sentences that may need revisions. I’m not a native speaker of English and the suggestions may not be the best. So, please consider referring to an English professional before taking action.

1.2 Introduction and background

The introduction section of the paper provides a good overview of the literature on automatic music composition. The authors described the limitation of the lack of structure and consistent style of automated music composition systems, which is the focus of this work. What I’m concerned about, however, is that the literature discussed in this paper is all in or before the year of 2019. For a more comprehensive background research, it would be necessary to include some reviews on more recent works, especially those related to modelling long-term dependencies & structural information in automatic music composition. For example, the following ones could be very relevant to the topic of this paper:
- Jin, C., Tie, Y., Bai, Y., Lv, X., & Liu, S. (2020). A style-specific music composition neural network. Neural Processing Letters, 52(3), 1893-1912.
- Chen, K., Zhang, W., Dubnov, S., Xia, G., & Li, W. (2019, January). The effect of explicit structure encoding of deep neural networks for symbolic music generation. In 2019 International Workshop on Multilayer Music Representation and Processing (MMRP) (pp. 77-84). IEEE.
- Dai, S., Jin, Z., Gomes, C., & Dannenberg, R. B. (2021). Controllable deep melody generation via hierarchical music structure representation. arXiv preprint arXiv:2109.00663.
- Zou, Y., Zou, P., Zhao, Y., Zhang, K., Zhang, R., & Wang, X. (2022, May). MELONS: generating melody with long-term structure using transformers and structure graph. In ICASSP 2022-2022 IEEE International Conference on Acoustics, Speech and Signal Processing (ICASSP) (pp. 191-195). IEEE.
In addition, there are some incomplete references (Line 489, 506, 507, 516, 517, 518, 523, 549).
In line 87, “Melody RNN”: Could you add a citation to the work?

1.3 Structure

The paper structure is slightly modified from the PeerJ standard sections. The Galician Xota section is friendly to readers who are new to the music style.

1.4 Figures

The figures are mostly well-designed, labelled, and referenced in the main context. I’ve got a few comments on Figures 1-3 (see below). Figures 7-9 look great!
- Figure 1: The figure could be improved by aligning the encoding with the MIDI visualisation along the time axis.
- Figure 2: In the figure caption, “where s, a and r represent” should be “where S, A and R represent” to keep consistent with the annotations.
- Figure 3: The figure caption says that the measure is 6/8. Could it also be 3/4?

1.5 Raw data

Raw data and code are provided following the PeerJ data policy.

Experimental design

2.1 Scope fitness

The paper describes the use of Double Deep Q Learning for automated composition of a specific music style – the Galician Xota. The topic of the paper – automated music composition – is well-suited to the scope of the PeerJ computer science journal.

2.2 Research question

The research question of this paper is well-defined, relevant, and meaningful. Following the limitation introduced in the background research, the work focused on investigating the use of Deep Q Learning for tunning the RNN-based composers towards a specific music style. What I would suggest is that it’s worth having a double-check on the knowledge gap after including some more recent literature review as I mentioned in “Basic reporting”.

2.3 Technical & ethical standards

The method and experiments described in this paper are a good start and the comparisons on different model configurations and RL rule-sets are well-defined. Still, there are some problems in the experimental design, which I would strongly suggest being revised before making the work public.
1. The dataset is spitted into 80% training and 20% validation, with no test set. In the experiments, the validation set is used for avoiding overfitting as well as for reporting the accuracies. This experimental design is not rigorous enough since the accuracies can be biased due to choosing the best model on the validation set during training. It would be necessary to rerun at least part of the experiments using a split including training, validation, and test set to ensure the validity of the results.
2. Regarding the evaluation, I would suggest providing the ground truth scores as a reference. According to the rule-set definitions in Table 2, the scores for ground truth music pieces in the dataset may not always be 100 (some of them may be lower, suggesting good performance can be achieved with a lower score). Providing the ground truth scores for a comparison could help in a better understanding of the model’s performance.
3. Apart from the objective evaluation reported in the paper, it would be helpful to also include a subjective evaluation. The rule-based evaluation can provide a good view of how good the compositions follow the musical rules. However, it cannot indicate how good the compositions are in general. A good score in Xota (or Xota+Megenta) rules does not always mean a good Xota composition.
Some more comments on the experimental details are included in the additional comments.

2.4 Sufficient details for replication

Sufficient details for reproduction are provided. It may be a bit hard to re-implement some of the complex rules by following the text description only, but readers can refer to the published code instead.

Validity of the findings

3.1 Results

The authors reported scores for all the rules listed in Table 1&2 as an evaluation of the compositions. A comprehensive comparison is made to investigate the effect of using different training data and different RL rule-sets. However, there are some problems in the experimental design which can make the results biased/insufficient. The problems are addressed in Section 2.3 of this review. Would you please revise the article following the related suggestions?

3.2 Conclusions

The paper suggested the success in composing the Galician Xota via using Double Deep Q learning and indicated the three important factors for good compositions related to rule design, which are the frequency of behaviour, the distance between action and reward, and the complexity of behaviour. The conclusions look good generally, the only comment I got is that please do include a subjective evaluation so as to validate the “success” in composing the Galician Xota. Since the current results can only indicate success in obeying the rules, which is not equal to a successful composition.

More detailed comments on the results and conclusions can be found in the additional comments.

Additional comments

Comments on clear, unambiguous, and professional English language:

- Line 19-20, “which immediately label the compositions as artificial.”: The sentence looks a bit strange. How about “which are hallmarks of artificial compositions.”?
- Line 20 & 24, “successful”: The word is unnecessary. It will be better to describe how successful the model is. For example, what is the performance improvement after using the proposed method for composing the Galician Xota music? For the second “successful” in line 24, is it describing the “implementation”? If yes, I would consider it as an unnecessary word since the “implementation” cannot be unsuccessful.
- Line 32, “they cannot truly be considered intelligent”: “they cannot be considered truly intelligent” or “they are not truly intelligent”.
- Line 35, “in the last years”: “over the past few years”?
- Line 39, “Variational Auto encoders”: It’s more common to use “Variational Autoencoders” or “Variational Auto-encoders”.
- Line 50, “RL Tuner”: “the RL Tuner”.
- Line 100-101, “From now on we will refer to these verses as section A for the volta and section B for the punto.”: It's a bit confusing, are "verse" and "punto" different things or not (as mentioned in the first sentence of this paragraph)? And are "these verses" the two verses mentioned in the previous sentence (the second is the repetition of the fist)? Or are they section A (the first) and section B (the second, repetition)?
- Line 111, “part A”: Is “part A” the same for “section A”, or are they different things? Same for “part B” in line 114.
- Line 119, “Major and minor”: Please keep consistent in using capitals.
- Line 161: Missing period.
- Line 201, “The state space S”: Here, the "S" has a different meaning from the "S" in Figure 1, which could be a bit confusing. It would probably help to use different acronyms so that readers can better distinguish between them.
- Line 217, “MDP”: Acronym not defined.
- Line 226, “rules-sets”: “rule-sets”.
- Line 232, “well established”: “well-established”.
- Line 333, 340 & 341: “M_Avg” & “X_Avg” should be “M_{Avg}” & “X_{Avg}”.

More detailed comments on experiments:

- Line 144, “an average of events per file of 860”: Could be rephrased to “an average of 860 events per file”. Are the events all MIDI events including time signature, key signature, and control changes? Or note-on and note-off events only?
- Line 145, “simultaneous tracks”: Are the tracks MIDI tracks? A single MIDI track can still be polyphonic. Are all the tracks in the dataset featured to be monophonic? And if not, how did you convert them into monophonic melodies?
- Line 146-147, “we separated the polyphonic pieces into… keeping only the tracks of substantial length”: How did you ensure the tracks kept have valid melodies? It could be possible that e.g. the music pieces are fluent in the original MIDI, but no longer fluent after keeping only one track.
- Line 205, “P_{ss’}^a”: What does ss’ and a mean?
- Line 210, reward function: Are the rewards equally weighted among all the rules? How is the score converted into the reward? Could you provide a formula for calculating the reward function R_{ss’}^a?
- Table 1, M1, Description, “C Major”: How about the transpositions after data augmentation?
- Table 1, M2, Description: How about ending with a tonic note followed by a rest?
- Table 1, X2: Is it a duplicate of the latter part of the M2 rule? If yes, is the model being rewarded twice when ending with the tonic?
- Table 1, X3, Description, “Excluding note-on or note-on events”: “note-on or note-off”?
- Table 1, XA3, Description, “in the beginning of section A: [0,0,0,0, ...].”: How about starting with longer rests? e.g. [0, 0, 0, 0, 0, 0, 0, 0, ....].
- Line 273, “the compositions were generated section by section”: Does it mean that the sections are independent of each other? If yes, do you have any mechanisms to model the inter-section dependencies?
- Line 311, “maximum validation accuracy of 78.27%”: How is the accuracy defined? Is it the raw accuracy, f-score, macro average accuracy, weighted average accuracy, rule-based score, or something else?

Detailed comments on the results and conclusion:

- Line 316, “single random note”: Is it possible to start with a rest?
- Line 324, “subsequently averaged”: Are they averaged among all sections, or among compositions of [A A B B A A] where sections A and B have different weights?
- Line 334-335, “suggesting that even a model… outperform the original Magenta RNN.”: It is not very precise, since it only outperforms the Magenta RNN in terms of following the rules. It could be possible that the compositions are worse.
- Line 357, “absolute performance”: How did you define absolute performance? The average score among all the rules? It would be better to describe the specific metric instead.
- Line 436, “we were able to accurately emulate the Galician Xota”: How accurately? With "accurately", I would expect an accuracy of at least >95%, which is not the case in this work.
- Line 442, “the most important factors”: Better to remove “the most” to avoid being absolute. The factors are important among the ones described in this paper under the defined metric, however, we cannot ensure there aren’t any other ones since there can be potential factors we didn’t discover.

Reviewer 2 ·

Basic reporting

The paper has a reasonable structure and a natural flow. Contextualising the study in the genre of Galician Xota (yes, we definitely need more diversity!), extending the RL Tuner work with the suggested methodology and the conclusions from various experimental settings are valuable contributions to the field. This study offers useful ideas to the researchers in the field of generative music (e.g. Galician Xota rule-set, “more general music rules not specific to the music style are important for developing style-specific models” etc.).

I think there are two main missing parts in the paper:

• Even though some musical files are provided in the supplementary material, there is no musical evaluation (by listening to the music) of the generated compositions in the paper. To be more explicit, yes, the paper contains a comprehensive analysis of the results within the scope of pre-defined analytically computable musical rules, but analysing musical compositions using only these rules is a limited way of valuing them. It would greatly add to the paper if some of the generated compositions are musically evaluated, even subjectively without any comprehensive human subject studies. (What do we actually hear in them? Is it possible for them to follow all of the rules, and yet stylistically and structurally be inconsistent?)

• Criticism of the 'style and structure' & the 'suggested methodology': The paper starts with the criticism of existing generative music models regarding their lack of style and structure, which is very valid in my opinion, but the suggested methodology that is limited to the defined set of musical rules does not convincingly suggest improvements in terms of the style and structure necessarily. In short, even though these rules are promising and sound reasonable, I think there is a need for further verification that these rules well capture the stylistic and structural properties in the style of Galician Xota. One way of increasing the confidence would be the above-mentioned point about the musical evaluation of the generated examples and checking the outstanding examples wrt the rules and their quality via listening tests (again, even subjectively, I think this still provides some information and a level of confidence for the other researchers).

Miscellaneous comments:

• I think more information about the Galician Xota should be provided earlier in the paper (how can we contextualise it?, which culture does it belong to?, one well-known example, a YouTube link etc.). It would be helpful to readers who are not familiar with the genre (and I think there would be a significant number of them).

• Since this paper focuses on stylistic and structural improvements for music generators in the context of Galician Xota, I think it would definitely be worth at least mentioning the transformer-based architectures (Music Transformer from Magenta, Perceiver AR etc.). For these exact same problems (style and structure), they have been promising in the field, and I think it’s worth mentioning them in the background section (and perhaps some problems with those attention-based architectures since they are not perfect).

• Line 93 typo (two ‘as’)

• Line 143: Since the duration of a MIDI file depends on the tempo, I think it makes more sense to provide number of notes, bars etc. information rather than providing an average time. Unless the tempo is static in the genre, I think it’s better to use bars, events, etc. in MIDI domain rather than actual time.

• Figure 4 is not clear.

• Some ablation studies with respect to the parameters such as reward scaler, discount factor etc. would help improving the paper as the values for these parameters are provided without any explicit justification.

• Line 340, 341: Subscript typo (AVG).

Experimental design

The architectures and techniques (LSTM-RNN + RL Tuner) used are reasonable and experiments are well designed within the scope of training corpora, rule-sets for the RL tuning part, and individual musical rules. The presentation of the results is diligent and informative. One interesting thing is that, in Figure 7, BasicRNN trained with the Xota dataset performs better than the BasicRNN trained with the Magenta dataset, with respect to M_avg (64% vs. 52%)? Does the Xota dataset better represent the Magenta rule-set than the Magenta dataset? I think it would be helpful to investigate the original Magenta training dataset and its characteristics.

Also, given a large amount of experimental data, they can be further explored focusing on individual and relatively more complex musical rules. The argument of “complex rules are generally more difficult to learn for the Deep Q Network (M5, M7, X4, and X5)” seems reasonable given the results and it would be a nice addition to elaborate more on these rules and try to come up with hypotheses explaining the model behaviours regarding them?

Validity of the findings

Within the scope of the experiments, the findings seem valid. Please see my comments above for the gap between ‘style and structure’ & ‘this methodology’ and in Experimental Design section.

Additional comments

This paper would make a great contribution to the field and it includes comprehensive experiments with valuable and reasonably convincing conclusions. I think considering the suggestions above would improve and add to the paper greatly. It would be too quick to make convincing conclusions about the stylistic and structural improvements of this approach, yet it is a promising direction. I think a musical analysis of the results (by actually listening to the generated compositions) is needed in the paper (even though there are too many generated samples, it would be helpful to focus on some outstanding ones, some good/bad examples given the rules). The literature on attention-based architectures for music generation would be very relevant here as they relate to this study, so referring to that literature could be helpful to this study.

---

## Round 0.2 · Major Revisions

The reviewers agree that the manuscript has been well improved - thank you. The remaining question is that of subjective evaluation by listener study.

Both R1 and R2 are correct to argue that including subjective evaluation would not be a problem for the paper's claim to be "scientific" or "reproducible". The authors perhaps have an understanding of reproducibility that is too narrow: for example, physical experiments (e.g. chemistry) may give slightly different results each time, but they can still be reproducible. PeerJ has no policy against user-based studies.

Both reviewers consider that rule-based evaluation alone does not give a complete evaluation of the system, in this particular case, and I concur. The reviewers' requested evaluation would make the outcome more convincing. R1 requests a user study, and R2 suggests subjective author evaluation may suffice. I agree that subjective listening comments are a minimum requirement, and a slightly more complete user study would make the paper even stronger. (This does not have to be difficult to run.)

Accordingly, I have recommended a decision of "Major Revisions". This is because PeerJ's "Minor Revisions" has a recommended turnaround of 10 days, which would not be enough time for subjective evaluations. I do not think that the requirements are major.

·

Basic reporting

Thank you to the authors for your updates and detailed replies to the comments! The paper has greatly improved in terms of language and reporting. The new paragraphs introducing the Xota style will be very useful to readers.

Experimental design

On the experimental design, firstly, apologies for my misunderstanding about the use of validation data in the initial review, it sounds reasonable in this case since the accuracies reported are for model tuning rather than the final evaluation.

Regarding the user study, I would (again) suggest having it in the evaluation. I personally do not agree with the reasons against a user study in the rebuttal:

For reason 1) regarding reproducibility, the reproducibility of this work can be achieved by having the comprehensive objective evaluation described in the paper. It's true that user studies are hard to reproduce. However, people do not usually expect user studies to be precisely reproducible and there are reasons why it is necessary. With a user study, people can better understand the music generation performance from a perspective view since it's always hard to evaluate music creation with a limited set of objective metrics (unless we can prove that the rules are reliable and complete enough for music generation evaluation).

For reason 2), a user study should not be limited to asking listeners how pleasing the created music sounds to them. It can also be useful to ask listeners to evaluate how well the model adapts to the Xota style. Adding a user study, in this case, will make the evaluation more complete and convincing. Besides, as reviewer 2 suggested, adding a listening test can greatly help in providing some evidence on how the method proposed is good at adding "style" and "structure" to music generation.

Besides, it's important to show that the model is generating *musical* outputs but not some patterns that adhere to the rules the model was trained with. This is especially necessary when the rules are used both for training as "music theory rewards" and for evaluation as the performance measurement.

Other related comments have been addressed in the updated version.

Validity of the findings

The related comments have been addressed in the updated version.

Additional comments

As mentioned in "Experimental design", I personally insist on having a user study, since having rule-based evaluation alone is not convincing enough. Still, it depends on to what extent PeerJ require open-source and reproducible research, in which case user study, indeed, has its disadvantage in reproducibility.

Reviewer 2 ·

Excellent Review

This review has been rated excellent by staff (in the top 15% of reviews)
EDITOR COMMENT
The reviewer was very clear in differentiating the assertions made by authors and reviewers, and in considering the balance of reproducibility versus subjectivity in evaluation. The written review has a clear but approachable tone, and I believe will be encouraging for the authors in improving their paper. Thanks!

Basic reporting

I think the authors have carefully considered the comments about reporting and improved the paper accordingly. Galician Xota genre has been better contextualised. All the typos and the structural comments are addressed and Figure 4 has been improved. Also, in the background part, transformers are mentioned as they are related to this work, following the reviewer comments.

Experimental design

My question was: ... (Given the better performance) Does the Xota dataset better represent the Magenta rule-set than the Magenta dataset? ...

The authors' answer: ... This does not necessarily imply that the Magenta dataset is better representative of the Magenta rules than the Xota dataset, although it is possible that this could be the case. ...

I think the answer is a bit unsound - Potentially, it could have been "This does not necessarily imply that the "Xota dataset" is better representative of the Magenta rules than the "Magenta dataset", although it is possible that this could be the case", given my concern.

The answer in the rebuttal says that indeed what's presented in the results is correct, which I truly trust in. Additionally, considering my question, even if the Xota dataset is better representative of the Magenta rules than the Magenta dataset, this is explainable as the Magenta rules are just basic music rules (not specific to any style), and maybe Xota dataset better represents them, which is fine. My suggestion was to further investigate this (which could improve the paper). It might not be feasible to further investigate given the Magenta dataset access that I understand. My concern is that the authors haven't fully understood the question. At this point my suggestion would be to include this arguably "question-raising finding" in the paper and provide a potential explanation, which can be a basic hypothesis.

Validity of the findings

I understand the authors claim that this paper is about defining a rule-set about a genre and using the rule-set as the guidance / success criteria, which is okay. And I see that the authors don't find much value in subjective evaluation in this context, but I think it would still be helpful to see " the authors' "subjective comments about the generated music in the paper for the following reasons. (if not a comprehensive subjective evaluation)

I understand the issue with the ground truth and the authors focus on the irreproducibility of the subjective tests, which is fair, but in my opinion, these don't make my question of "What do we actually hear in them?" any irrelevant or less valuable. So, even though it's not reproducible, I would still love to learn about "what they do hear in this music", as they have spent a considerable amount of time with the system and they are better experts in Galician Xota than me (and potentially more than many of the readers). Basically, the motivation for that question is to convey their experience to the others in the community in a more efficient and compact way, which would be helpful to the other practitioners who would like to use or improve this system. Obviously, the other practitioners don't have to agree with their subjective comments, but even simple insights about the stylistic closeness of the generated pieces to the Galician Xota genre and some musical observations would be helpful.

Including subjective comments (if not a comprehensive subjective evaluation) is an opportunity to raise our confidence in the Xota rule-set. The approach of defining a rule set and building a system that can generate music that satisfies those rules is questionable by itself, especially under such a stylistic claim. How do we know that the rule-set is meaningful, and more importantly, it represents the Galician Xota genre? Yes, I'm sure the musicological analysis is supportive, but it's also hard to define any genre with tens of explicit rules. They are helpful, but inherently questionable. So, considering all the points above, I still suggest that the authors' subjective comments would be insightful for the reader.

---

## Round 0.3 · Minor Revisions

The paper is now looking good, almost ready. I request that you address the final small issues raised by both reviewers. I think this can be handled very quickly.

·

Basic reporting

Related comments have been addressed in the new version.

Experimental design

Many thanks for the user study update! It will be very useful to work as a subjective evaluation of the model's ability in generating music in the Xota style. The user study looks pretty nice and will be a good addition to the paper's completeness. I just have one comment related.

Since the section As and Bs are generated separately, I assume one whole piece of the generated Xota music will be combined with 6 generated sections (i.e. in the [A A B B A A] structure). If this is the case, how are the six sections (or two if the As and Bs are the same ones) dependent on each other? If they are generated separately, is there any evaluation of the coordination of the generated music piece? Also, since the A and B sections are generated using different models (trained on the Xota rules of A and B, respectively), how reliable is the "Structure" evaluation in the user study? Could you provide some insights on why the Magenta rule set can be so helpful in terms of the "Structure" evaluation? Is the improvement introduced by the Magenta rule set, or is it simply because the As and Bs are trained using two sets of data?

Validity of the findings

Related comments have been addressed in the new version.

Reviewer 2 ·

Basic reporting

I thank the authors for their careful consideration of the feedback.

Experimental design

It is great that they included the finding about the Xota dataset / Magenta dataset with a hypothesis, which I think has improved the paper. The argument of "Furthermore, it is composed of thousands of different MIDIs from different genres which may not follow the general rules set out in the Magenta rule-set as consistently as the pieces in the Xota dataset, which were carefully curated for the purposes of this work." is not necessarily convincing to my understanding: Why would carefully curating a Xota dataset for this work would lead to satisfying the generic musical rules presented in the Magenta rule-set more than the Magenta dataset itself? One more question, why would having many different genres go against following generic musical rules in the Magenta set? This would make us question the generic-ness of the rules. In short, even though an in-depth analysis is not necessary within the scope of the paper, if there is a chance to improve the paper, I would suggest authors to be careful about these arguments.

Also, I think adding the subjective evaluation section has improved the paper, thanks to authors for their consideration. It is reasonable and a good thing to have an expert musicologist in the genre as the genre is very niche and I think it's fine if they are one of the authors (if this is a concern at all). If there is an opportunity to improve the paper before the final version, I would suggest the following regarding the subjective evaluation section. To my understanding, the subjective comments are still technically oriented focusing on melody, rhythm, structure etc. which shouldn't be the case necessarily. They are helpful, but it would be also helpful to know about the cultural relevance of the generated pieces to the Galician Xota genre beyond characterisation, for instance, do they remind you of any composer or any piece from the tradition? Any high-level patterns or tendencies observed in the generated pieces beyond the metrics? Given the advantage of having a musicologist in the team, I think these could improve this section.

Validity of the findings

Please see my points above.

Additional comments

I think the authors have carefully considered the main points I've suggested earlier, thanks for that. If there is a chance to improve the paper before the final version, I would suggest being careful about my points above.

---

## Round 0.4 · accepted · Accept

In my judgement, your changes have adequately addressed the remaining feedback. Thank you.